# High prevalence of focal and multi-focal somatic genetic variants in the human brain

Michael J. Keogh[1], Wei Wei[1], Juvid Aryaman[2], Lauren Walker[3], Jelle van den Ameele[1], Jon Coxhead[4], Ian Wilson[4], Matthew Bashton[5], Jon Beck[6], John West[6], Richard Chen[6], Christian Haudenschild[6], Gabor Bartha[6], Shujun Luo[6], Chris M. Morris[3], Nick S. Jones[2,7], Johannes Attems[3] & Patrick F. Chinnery[1,8]

Somatic mutations during stem cell division are responsible for several cancers. In principle, a similar process could occur during the intense cell proliferation accompanying human brain development, leading to the accumulation of regionally distributed foci of mutations. Using dual platform >5000-fold depth sequencing of 102 genes in 173 adult human brain samples, we detect and validate somatic mutations in 27 of 54 brains. Using a mathematical model of neurodevelopment and approximate Bayesian inference, we predict that macroscopic islands of pathologically mutated neurons are likely to be common in the general population. The detected mutation spectrum also includes *DNMT3A* and *TET2* which are likely to have originated from blood cell lineages. Together, these findings establish developmental mutagenesis as a potential mechanism for neurodegenerative disorders, and provide a novel mechanism for the regional onset and focal pathology in sporadic cases.

[1] Department of Clinical Neurosciences, University of Cambridge, Cambridge Biomedical Campus, Cambridge CB2 0QQ, UK. [2] Department of Mathematics, Imperial College London, London SW7 2AZ, UK. [3] Institute of Neuroscience, Newcastle University, Campus for Aging and Vitality, Newcastle upon Tyne NE4 5PL, UK. [4] Institute of Genetic Medicine, Central Parkway, Newcastle University, Newcastle Upon Tyne NE1 3BZ, UK. [5] Wolfson Childhood Cancer Research Centre, Northern Institute for Cancer Research, Newcastle University, Newcastle Upon Tyne NE1 7RU, UK. [6] Personalis Inc, 1330 O'Brien Dr, Menlo Park, CA 94025, USA. [7] EPSRC Centre for Mathematics of Precision Healthcare, Imperial College London, London SW7 2AZ, UK. [8] MRC Mitochondrial Biology Unit, University of Cambridge, Cambridge CB2 0XY, UK. These authors contributed equally: Michael J. Keogh, Wei Wei. Correspondence and requests for materials should be addressed to P.F.C. (email: pfc25@cam.ac.uk)

Common neurodegenerative disorders, including Parkinson's disease (PD) and Alzheimer's disease (AD), are characterised by toxic protein aggregation and cell loss in defined brain regions[1,2]. The majority of patients have no family history, but in ~5% of cases, germ line genetic variants in one of ~50 genes either cause or contribute to disease risk, with a clinically indistinguishable phenotype[3,4]. For most neurodegenerative disease genes, a single mutated allele is required to cause disease through haploinsufficiency or a dominant negative effect. This raises the possibility that somatic mutations arising in the same genes within a specific cell lineage contribute to the pathogenesis of non-familial cases. Although we have no direct evidence, islands of cells containing these mutations could synthesise misfolded proteins with the potential to spread throughout the brain during human life[5]. Technological limitations have prevented this question from being addressed, but if correct, then developmental mutagenesis could be a major cause of sporadic neurodegenerative diseases.

The hypothesis can be tested by sequencing a very large number of single neurons, or by ultra high-depth re-sequencing DNA extracted from a pool of cells isolated from brain tissue samples. Since the overall low frequency of somatic mutation events approximates the intrinsic DNA sequencing error rate, some form of experimental validation is critical. This is not possible with single-cell approaches because individual neurons and glia are destroyed during the sequencing process. Although recent single-cell studies provide evidence that somatic mutations do occur in the developing brain[6,7], understanding the total mutational burden across the human brain will require a massive sequencing effort. Here we took a complementary approach, harnessing ultra-high depth sequencing to survey the somatic mutations across different brain regions. Using a computational model of brain development, we extrapolate our findings across the human brain, and provide an estimate of the somatic mutation burden in the human population.

## Results and Discussion

**Performance metrics and variant validation.** We sequenced all exons of 56 genes known to cause, or predispose to, common neurodegenerative disorders (132,617 base pairs (bp)) (Supplementary Table 1, left), and 46 control genes expressed at low levels in the brain which are typically associated with cancer (152,519 bp) (Supplementary Table 1, right, subsequently referred to as 'cancer' genes). The sequencing pipeline was developed to ensure that 99.6% of coding bases were covered at >1000-fold depth across both gene panels, some of which are notoriously difficult to sequence comprehensively (e.g., *MAPT* in Fig. 1a and Supplementary Fig. 1). Spiked HapMap control samples called by differing combinations of variant callers enabled us to establish a variant calling algorithm with a high sensitivity and specificity (Fig. 1b–d and Supplementary Fig. 2, see Methods). Somatic mutations that were present in only one brain region (subsequently termed single region mutations, SRMs) and those that were present in more than one sample (multi-region mutations, MRMs) above 0.5% variant allele frequency (VAF) (Fig. 1b, c) were optimally called by utilising the high sensitivity afforded by MuTect2[8] or Varscan2[9,10], followed by deepSNV[11,12] to ensure high specificity confirmation calling which takes into account base error rates within the platform (Fig. 1e and Supplementary Fig. 2, see Methods). This resulted in a calling pipeline with 93% sensitivity and 99% specificity to detect variants above a VAF of 0.5% (Fig. 1b and Supplementary Table 2). A greater proportion of the HapMap variants at low VAF were detected when the sequence depth increased from 1000-fold to 7000-fold. VAF > 1% were consistently detected when the sequencing depth was

>1000-fold, but VAF >0.5% required >4000-fold depth to minimise the false-negative rate of any caller across the 102 gene 285 kb panel (Fig. 1d).

Ultra-high-depth re-sequencing was performed on 173 frozen brain regions from post-mortem cases of AD ($n = 20$ brains), Lewy body (LB) disease (PD or Dementia with LB: $n = 20$ brains), and age matched controls with no significant neuropathology ($n = 14$ brains) (Supplementary Tables 3 and 4). Paired blood DNA was available for sequencing in six subjects (control: $n = 2$, AD: $n = 1$, LB: $n = 3$). Quantitative immunohistochemistry of hyperphosphorylated tau (HP-T), β-amyloid, and α-synuclein was performed on all selected brain regions (cerebellum: CB = 54, Entorhinal cortex: EC = 53, Frontal cortex: FC = 32, Medulla: Med = 24, Cingulate: Cin = 10) (Supplementary Tables 3 and 4). An overall mean sequencing depth for the 102 genes using the accuracy and content enhanced (ACE) platform of 5374-fold (s.d. = 745) (Fig. 1f) was achieved, set in order to minimise the false-negative rate determined from HapMap-spiked control samples (Fig. 1d). Independent validation of the detected variants was performed on the same tissue samples using an orthogonal technique, which incorporated barcode-labelled amplicons (Haloplex[HS], sequenced to a mean depth of 6830-fold, s.d. = 1549) (Fig. 1f). After quality control (QC), the ACE platform detected 62 somatic variants (56 single-nucleotide variants, SNVs; and 6 insertion-deletion variants, indels) in 44 brain samples (50% of the entire cohort of brains—controls: $n = 6/14$, AD: $n = 9/20$, LB: $n = 12/20$) and 4 blood samples. Totally, 56 of these variants were also covered by the Haloplex[HS] platform, showing a strong concordance in the VAF measured by the discovery and validation platforms ($r^2 = 0.953$, $P < 2.2 \times 10^{-16}$) (Fig. 2a, b).

**Single regional mutations.** Eighteen somatic mutations (56.4% of the total 39 detected variants) were present in only 1 brain region within an individual, and 4 somatic variants in only 1 blood sample (Fig. 2c, d, Fig. 3a–e, h, Supplementary Figs. 3–5 and Supplementary Table 5 and Supplementary Data 1). These SRMs were detected at a mean VAF of 0.84% (s.d. = 0.005). Confidence intervals for the VAF were based on the normal approximation to account for any potential sampling bias arising during sequencing (Fig. 3a–d). The SRMs were equally likely to arise in any brain region (CB = 6/54 brains, 11.1%; EC = 7/53 brains, 13.2%; FC = 2/32 brains, 6.3%; Med = 3/24 brains, 12.5%) and were equally likely to occur in neurodegenerative disease (7/ 132,617 bp) or cancer genes (11/152,519 bp) ($P = 0.64$, Chi-squared with Yates correction) (Supplementary Figs. 3–5). The majority of SRMs in brain were C > T substitutions ($n = 14/ 18$, 77.8%) (Fig. 3f, h) consistent with spontaneous deamination of 5-methyl-cytosine[13], as observed in single neurons[14]. Purine–purine transitions on the non-template strand were exclusively seen in case genes ($n = 4/7$) ($P = 0.01$ vs. Control genes, Fisher's exact test) in keeping with replication-transcription collisions[15], as seen in single cortical neurons[14]. However, given that the same rare mutations were detected in ≥0.5% of the mapped reads, and were therefore present in many cells, it is highly likely that they arose during development. The flanking 5′ and 3′ sequence of the SRMs were distinct from mutational 'signatures' described in cancers[16], suggestive of a different mechanism of mutagenesis (Fig. 3f and Supplementary Table 6). Seven mutations occurred in neurodegenerative disease genes with a mean VAF of 0.82% (s.d. = 0.003, range: 0.47–1.56%). There was no difference in the proportion of SRMs in neurodegenerative disease genes between any disease group (AD 5/20; LB: 1/20; control 1/14) (Fig. 3a–d). Based upon the observed frequency of SRMs in neurodegenerative disease genes in this study (13% of brains), we estimate that 190 cases and 190 controls would be required to detect a twofold increase in the

incidence of SRMs in a disease group with >90% power (Supplementary Fig. 6).

Using a simplified mathematical model of neurodevelopment to assist with the interpretation of our experimental data, we modelled brain development as a deterministic branching process, based on the assumption that de novo mutations are randomly distributed to daughter cells[17], and that symmetric cell divisions are common throughout neurodevelopment. We also studied the potential impact of a simple model of cell death during brain development (ranging from 50%[18,19] to >99%), finding that this had negligible impact on the qualitative behaviour of the model (Supplementary Note 1).

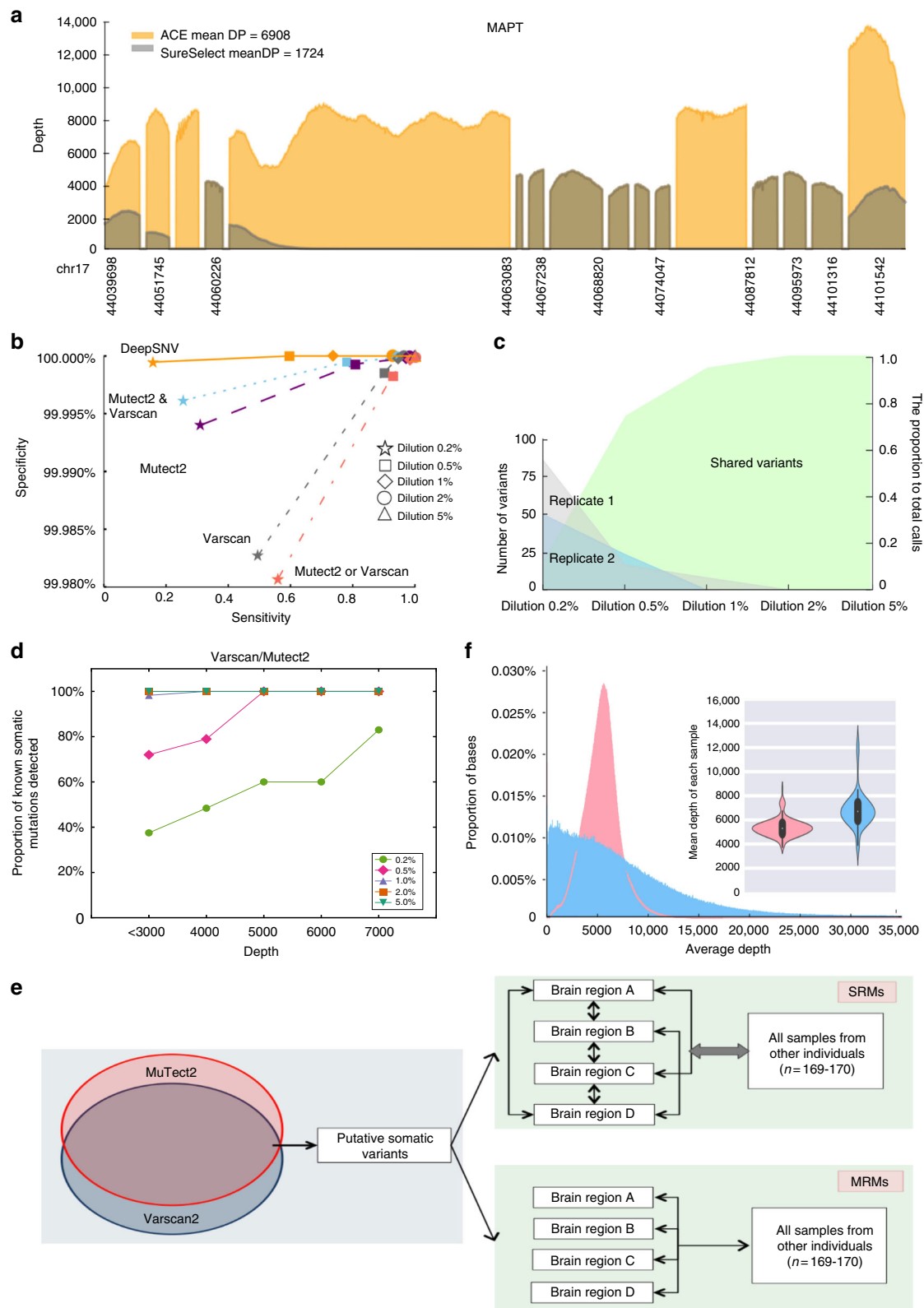

The Haloplex[HS] barcode tagging approach allowed us to determine both the number of cells sequenced (~611,285), and the proportion of cells containing a somatic mutation (Fig. 4a, b). We determined the somatic focal mutation rate in the human brain to be $4.8 \times 10^{-10}$–$2.99 \times 10^{-9}$ per base per cell division (95% Bayesian credible interval (BCI)) (Fig. 5a, b; Supplementary Methods). This closely approximates the somatic mutation rates in mitotically active tissues[20], providing further reassurance that our model was a reasonable model of cell division in the developing human brain. We then simulated the neurodevelopment of individuals under the model, given the inferred mutation rate, to find the frequency and size of brain regions harbouring known pathological mutations in neurodegenerative disease genes (Human Gene Mutation Database[21]) (Fig. 5c–h). We found that each individual tended to have $10^5$–$10^6$ pathologically mutated cells (Fig. 5e). Similar results were obtained using an alternative modelling approach, where the brain was assumed to consist of regions of a fixed size, each with an independent probability of being homogeneously mutated (see Supplementary Note 6, Supplementary Fig. 11E). We found that 10.8% (±0.1% s.d.) of simulated individuals possessed one or more regions of $2.62 \times 10^5$ spatially contiguous pathologically mutated cells (Fig. 5f). Furthermore, each individual brain was found to harbour 75–481 regions (95% BCI) of 128 pathologically mutated cells (Fig. 5g).

Our modelling approach is a crude approximation of real neurodevelopment, neglecting subtleties relating to the precise mechanisms of cell death; the existence of a progenitor founder pool; asymmetric cell division in the later stages and cellular migration, for example. In Supplementary Notes 1–6, we explore several alternative models which attempt to address these complexities (Supplementary Figs. 8–11). Overall, we find that many of our conclusions are broadly robust to these alternative model structures. The central conclusion from these models is that if neurodevelopment is topologically similar, but not necessarily equivalent, to a deterministic branching process, then we expect islands of pathologically mutated cells to exist. Furthermore, if we assume that during neurodevelopment the mutation rate of DNA is on the order of $10^{-10}$–$10^{-9}$ mutations per base per cell division[20,22] (Fig. 5b), given unbiased, spatially proximal, replication of daughter cells once the brain consists of approximately $10^6$ cells, then a simple order-of-magnitude estimate (Supplementary Note 4) suggests that every individual is expected to possess 1 pathologically mutated focal mutation consisting of approximately $10^4$–$10^5$ cells once the brain has fully developed. This argument is independent of the neurodevelopmental mechanism prior to the brain consisting of $10^6$ cells. Our model serves as an initial means to explore the prevalence of these mutated foci, suggesting that such foci are possessed by almost all individuals. These regions may have the potential to generate mutant proteins that form novel fibrillar structures, which could spread and cause different neurodegenerative diseases[23], or modify the clinical phenotype, depending on the original mutated gene.

**Multiple regional mutations**. Seventeen mutations (43.6% of the total 39 detected variants) were present in more than one brain region, or in a paired blood sample and brain (Fig. 2c, d, Fig. 3a–e, h, Supplementary Table 5, Supplementary Figs. 3–5 and Supplementary Data 1). These mutations had a significantly higher VAF than SRMs (3.67%, s.d. = 0.04, $P = 0.0024$) (Fig. 3a–e). Only one of these variants occurred in a neurodegenerative disease gene (Case number 12: p.R464R in *TAF15*, mean VAF 6.23%, s.d. = 0.016), which was present in all 3 brain regions sampled from this single control individual (VAFs in cerebellum: 4.37%, entorhinal cortex: 6.98% and frontal cortex:7.35%). Based on these observations, we estimate that mutations within neurodegenerative disease genes will be present diffusely across the brain in up to 9.77% of all humans (95% CI: 0.33%–9.77%, Wilson score interval test). Sixteen MRMs (94%) occurred in cancer genes, with 15 (93.8%) known to be associated with myeloproliferative blood disorders ($n = 19$ of the 46 genes in the cancer panel). This was greater than expected when compared to solid organ tumour or non-cancerous control genes (Supplementary Table 1) ($n = 27$ of 46 genes, $P = 7.45 \times 10^{-6}$), raising the suspicion that specific MRMs were derived from the circulating blood cells. The two genes most frequently mutated in our study (*DNMT3A*, $n = 6$; *TET2*, $n = 6$) account for the majority of age-related clonal haematopoietic mutations[24,25]; and four of the MRMs (23.6% of MRMs, *DNMT3A* p.R882H, *DNMT3A* p.P700L, *DNMT3A* c.1667_splice, *TET2* c.3472_splice) involved known mutational hotspots[25] (Supplementary Data 1). Given that these specific alleles were also detected in our study, this strongly supports a clonal hematopoetic origin for these particular mutations rather than an early developmental origin within the nervous system. In keeping with this, the VAF of the myeloproliferative gene mutations was always greater in available paired blood samples than in the brain ($n = 4$) (Figs. 2c, Fig. 3e and Supplementary Figs. 3–5). However, the fold difference was surprisingly low (mean blood:brain VAF ratio = 7.92, range: 2.10–11.98), suggesting that at least some of the rare clonal mutations were present in cells outside the vasculature, most probably including migratory immune cells[26].

**Fig. 1** Genotyping platform performance and quality control. **a** Coverage plot of the *MAPT* gene on the ACE platform highlighting the augmented coverage over and above that seen within the SureSelect alone (yellow—regions covered by custom augmented probes, brown regions—covered by ACE probes without augmentation). **b** Sensitivity and specificity of 5 combinations of variant callers at different VAF of HapMap control mixes. DNA from cell-line NA12878 was spiked into DNA from cell-line NA12877 using 2, 5, 10, 20, or 50 ng of NA12878, making up to 1 μg total DNA with NA12877 to create relative VAF of 0.2%–5%. deepSNV, MuTect2, Varscan2 somatic caller and Varscan un-paired calling were employed in different combinations to determine the sensitivity and specificity to detect variants at each VAF. The optimum caller pipeline was set at a VAF of 0.5% using a dual calling approach for variants called by either MuTect2 or Varscan2, which had a 92.98% sensitivity and 99.9984% specificity. **c** Number of observed variants called by either MuTect2 or Varscan2 at each VAF in biological replicates highlighting consistent performance of the sequencing pipeline. **d** Proportion of known variants present in the mixed HapMap controls that were detected using our primary calling pipeline. VAF > 1% were consistently detected when the sequencing depth was >1000-fold, but VAF > 0.5% required >4000-fold depth to minimise the false-negative rate of any caller across the 102 gene 285 kb panel. **e** Schematic overview of the calling algorithm used in the study. Variants called from paired sample calling by either MuTect2 or Varscan2 were selected. All brain regions from those individuals were then compared to both each other (for intra-regional variation), and then to all other individuals (for inter individual variation) for those alleles to ensure that detected variants from Varscan2 or MuTect2 were truly focal or Multi-focal in nature. **f** The average depth per base for each sequencing platform: ACE (pink) or Haloplex[HS] (blue). The mean sequencing depth across the whole panel per sample is shown in the inset violin plot for both platforms again using the same colour scheme

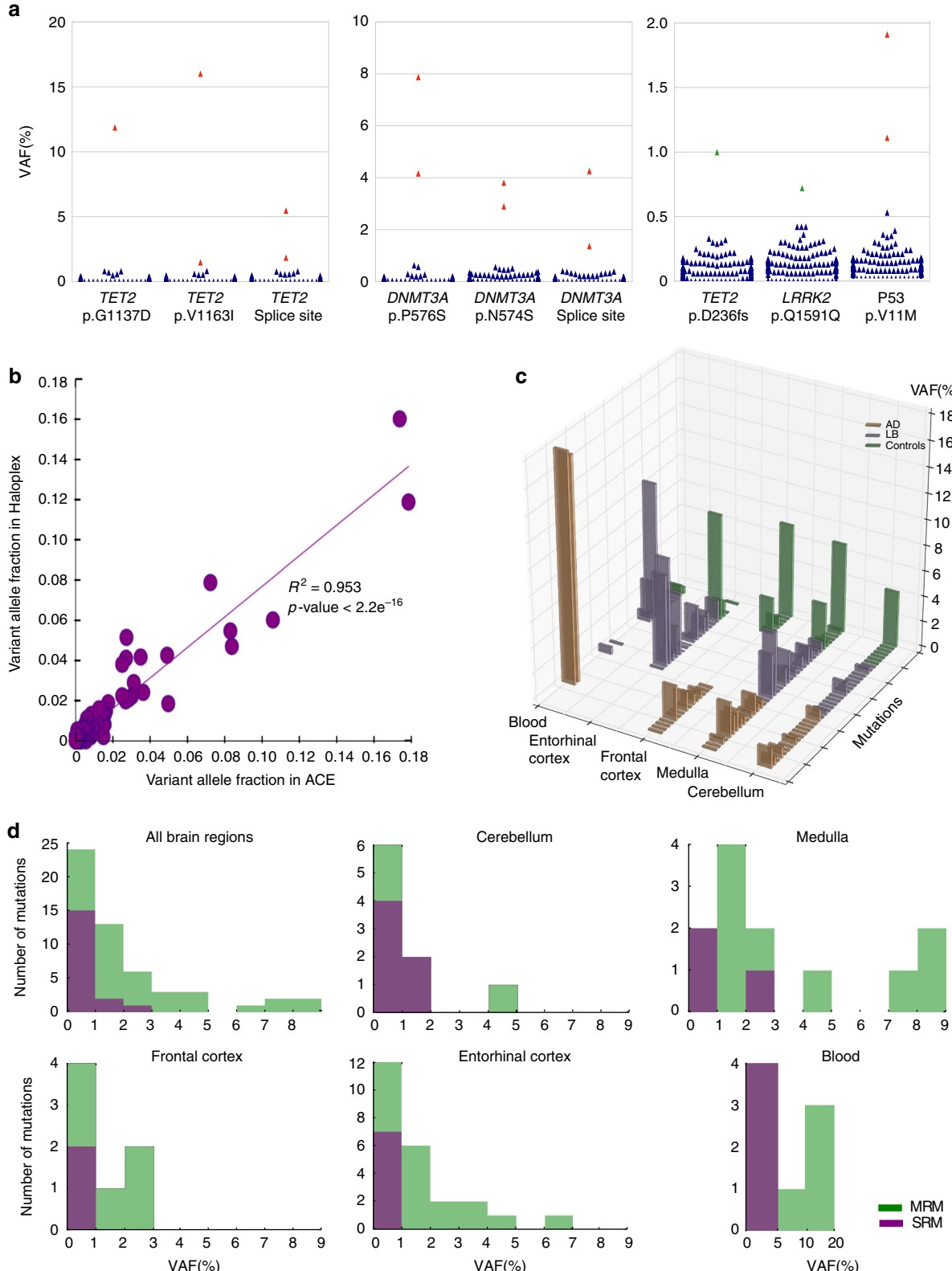

**Brains with more than one mutation**. Nine brains possessed more than one mutation across all samples sequenced (Supplementary Table 5, Supplementary Figs. 3–5 and Supplementary Data 1). Four of these brains had two mutations in genes associated with haematological malignancies which were either detected in blood or showed the same profile in different brain regions, thereby consistent with translocation into the brain. However, three individuals had a mutation profile that may cast light on the development of somatic mutations within the brain. Brain 17 (AD) had two focal mutations within the cerebellum (*UCHL1* p.I93I (1.56% VAF), and *PDGFRB* p.I985fs (0.64% VAF)). Whilst the detection of more than one mutation within a

**Fig. 2** Somatic mutations detected in 179 samples from 54 human brains. **a** Examples of somatic mutations initially detected using SureSelect ACE platform and then validated on the Haloplex[HS] platform. Each plot shows VAF across all the brain samples for the specific alleles indicated. Somatic mutations with VAFs > 10% are shown in the left plot, somatic mutations with 2% < VAFs < 10% are shown in the left and middle plots, and somatic mutations with 0.5% < VAFs < 2% are shown in the middle and right plots. SRMs identified by the SureSelect ACE platform are shown in green, and MRMs are shown in red. Blue symbols are the VAF which fall below the threshold for detection on the Haloplex platform. Thus, the data show that variants detected by SureSelect ACE are also detected by the Haloplex[HS] platform, which has a negligible false positive rate for VAF > 0.5%. **b** Correlation of VAF for the 56 detected alleles in the ACE platform and the Haloplex[HS] platform. **c** VAF in each tissue type / brain region for each of the 39 detected somatic mutations. These mutations are also co-coded by the disease cohort of the case in which they were detected. **d** VAF of each mutation (SRM: purple, or MRM: green) within each tissue. This shows a predominance for low VAF particularly for SRMs in all tissues, with higher VAFs being observed for MRMs

single tissue sample could reflect the stepwise accumulation of somatic mutations, with competition between subclones and sequential subclone evolution[27−29], being synonymous, the *UCHL1* variant itself is unlikely to have promoted the evolution of the *PDGFRB* mutation (Supplementary Fig. 3). Brain 34 (AD) contained two focal splice-site mutations in *SETX* and *ERBB2* within the entorhinal cortex. *ERBB2* encodes the erbB2 proto-oncogene and splice variants can act as major oncogenic drivers[30], which potentially increased the likelihood of developing other somatic mutation such as that observed in *SETX* (Supplementary Fig. 3). However, against this, the two variants had similar VAFs (*SETX*: 0.60% and *ERBB2*: 0.71%), suggesting that they are independent rare events in the pool of sequenced cells. Finally, in brain 54 (LB) we observed two mutations in *TP53* (Supplementary Table 5, Supplementary Fig. 5 and Supplementary Data 1): p.V143M was found in the medulla (VAF 1.75%), and entorhinal cortex (VAF 0.86%); and p.R174W was present only in the entorhinal cortex (VAF 0.46%). This is consistent with the hypothesis that p.V143M arose earlier in brain development before the p.R174W mutation arose within a subclone of cells only populating the entorhinal cortex. Given the strong pro-oncogenic effects of *TP53* gene mutations[31], these data also provide a possible explanation for unexplained brain malignancies[32]. Long-read length deep sequencing will be required to resolve these possibilities.

**Potential associations with disease.** Finally, and intriguingly, non-synonymous or frame-shift mutations in hematopoietic disorder genes were detected in 40% of LB brains (8/20), in contrast with controls (7%, 1/14; $P = 0.05$, where the frequency was consistent with previous reports[24,25,33]). Given the role of *DNMT3A* and *TET2* in regulating DNA methylation[34,35], these findings provide one explanation for the concordant changes in DNA methylation seen in the blood and brains of PD patients who share LB pathology[36]. The spatial distribution of *DNMT3A* and *TET2* variants mirrored the quantitative neuropathology, with the VAF in medulla 2.1-fold greater than entorhinal cortex (s.d. = 0.69, $P = 0.0064$). This could reflect regional weakness of the blood-brain barrier seen in PD[37] and other neurodegenerative disorders[38]. Given that clonal haematopoetic variants in *TET2* accelerate age-related atherosclerosis in mice[39], our findings raise the possibility that blood cell precursors harbouring somatic mutations translocate into the brain and contribute to the pathogenesis and clinical presentation of neurodegenerative diseases through cells derived from myeloid precursors[40].

In conclusion, based on observations from 173 human brain tissue samples, and ~611,000 cells, our findings indicate that the human brain is highly likely to contain many zones of cells harbouring somatic mutations, including mutations affecting neurodegenerative disease genes. Our extrapolations are based on the assumption that there are similar rates of mutation across the genome, which we accept may not be the case. However, if these mutations involve neurodegenerative disease genes, they could contribute to the pathogenesis of neurodegenerative diseases.

Although pathogenic mutations occurring within the first ~1000 cells during brain development are rare, these are likely to cause a disease phenotype because they will affect a large proportion of brain neurons. Our study was not sufficiently large to show this directly, but this provides a potential explanation for common sporadic neurodegenerative diseases which currently affect ~10% of people in the developed world[41,42]. It is conceivable that detecting these mutations during life will increase diagnostic precision, leading to new therapies, particularly if they involve targets amenable to pharmacological intervention within vulnerable neural circuits[43].

## Methods

**Post-mortem brains and histopathology.** Frozen brains were identified from the Newcastle Brain Tissue Resource (NBTR) fulfilling both pre and post-mortem criteria for either AD ($n = 20$); PD dementia or Dementia with Lewy bodies ($n = 20$); and healthy controls >65 years old with no clinical ante mortem history of cognitive impairment or movement disorder, no family history (>1 first degree relative) with any neurodegenerative disease, and neuropathological features consistent with normal aging ($n = 15$) (Supplementary Table 3). Samples were provided following ethical approval from Newcastle University Brain Tissue Resource (NBTR2013083PC). Totally, 54 cases remained after QC of the sequencing data (Supplementary Table 3) and 1 cm$^3$ blocks of grey matter were carefully manually dissected from each region (frontal cortex, entorhinal cortex, cingulate gyrus and medulla) after excluding macroscopically identifiable white matter or vascular tissue (Supplementary Tables 3 and 4).

Quantitative neuropathological data was obtained on fixed sections from the same regions using AT8 (for phospho-tau), 4G8 (for β-amyloid) and α-synuclein antibody staining on slides from all brain regions except the cerebellum. The relative quantity of the proportion of area covered by immunopositive staining was performed as previously described[44], and after correction for white-matter area where appropriate (Supplementary Table 4).

**Sequencing and validation platforms.** Two panels were defined incorporating 102 genes (Supplementary Table 1). Panel 1 (132,617 bp): 56 genes known to cause monogenic forms of neurodegenerative disease or disease risk modifiers (odds ratio > 2) identified through a systematic review[45]. Panel 2 (152,519 base-pairs): 46 genes associated with cancer selected from an existing clinical cancer panel (Personalis Inc., USA) showing the lowest transcript levels in the brain. Primary genotyping was performed using the ACE technology (Personalis Inc., USA) containing a backbone of the SureSelect probe set (Agilent, USA) with additional custom capture probes to significantly improve overall coverage ACE platform, in which the augmented probe design enabled coverage of 99.6% of coding regions at >1000-fold depth (Supplementary Fig. 1). An example of this augmented coverage in *MAPT* is shown in Fig. 1 and *TET2* and *DNMT3A* in Supplementary Figure 1b. Validation was performed using Haloplex[HS] capture technology designed using the Agilent SureDesign tool to capture all exons and 25 bp of intronic flanking regions of all 102 genes (Supplementary Fig. 1).

For the ACE platform, 1 μg of genomic DNA was extracted from brain, blood or HapMap controls, and was Covaris sheared, end repaired and ligated with adaptors. Adaptor ligated DNA fragments were amplified by PCR with 6 cycles, and subjected to SureSelect enrichment with probe panels for all 102 genes. An additional 10 cycles of PCR were performed with enriched material. For the Haloplex[HS] platform, in parallel, but independently, equal quantities of DNA to those utilised for the ACE platform were enriched by the same method. Sequencing of enriched DNA from both capture protocols was performed using HiSeq 2500 (Illumina, San Diego, CA, USA) sequencers with single lane, paired-end 2 × 101 bp reads and Illumina's proprietary Reversible Terminator Chemistry (v3).

**Sample dilutions for the level of detection testing.** The limits of detection of minor alleles was determined for both platforms using HapMap CEPH cell-line DNA (Coriell Institute) to simulate allele frequencies (AF) ranging 0.2, 0.5, 1, 2 and

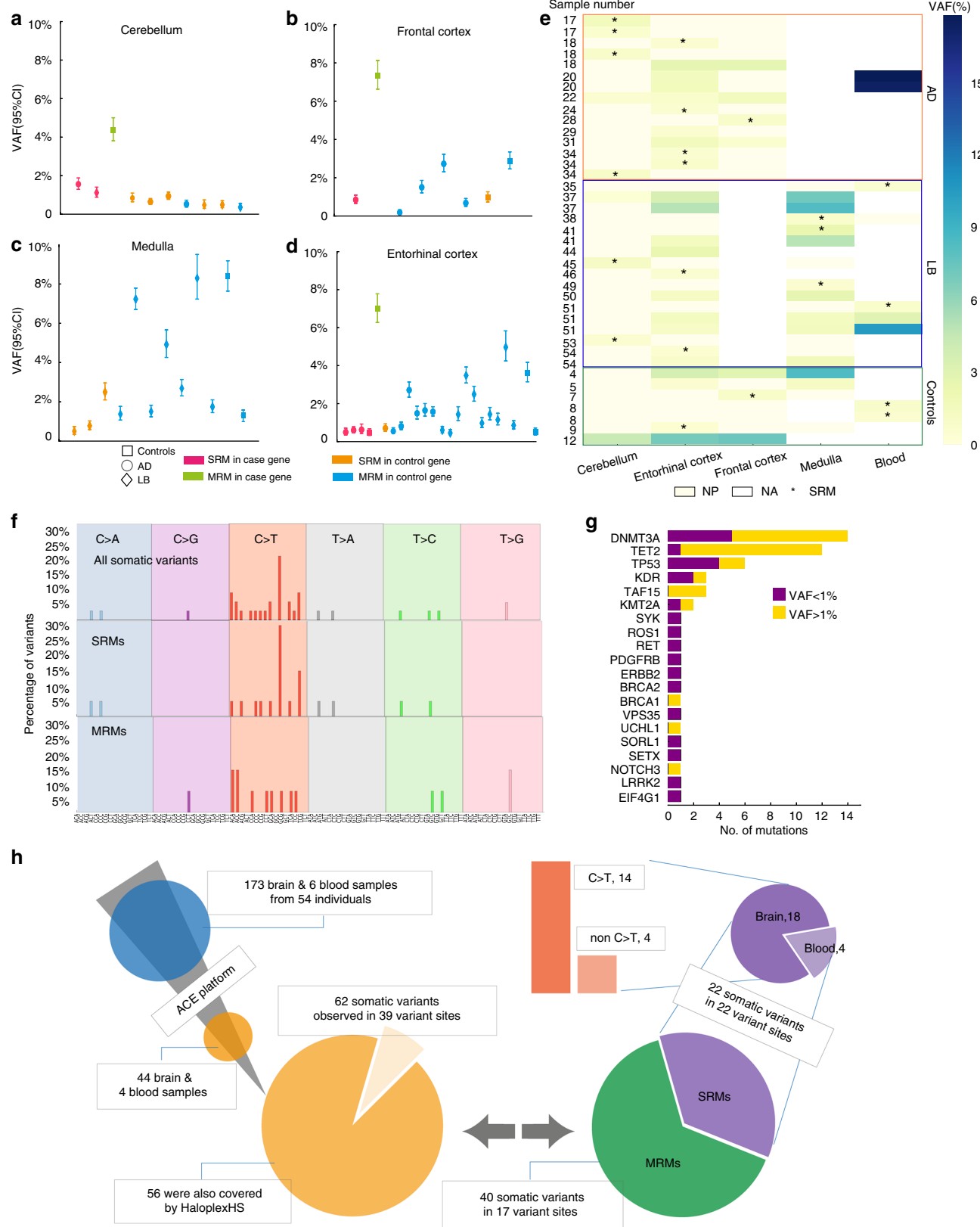

5%. DNA from cell-line NA12878 was spiked into DNA from cell-line NA12877 by using 2, 5, 10, 20 or 50 ng of NA12878, making up to 1 μg total DNA with NA12877 thus achieving the stated simulated VAFs. High confidence, pedigree consistent germline variants, concordantly called on several platforms from high-depth after PCR-Free sample preparation were obtained for each sample from

http://www.illumina.com/platinumgenomes/, and sequencing reads were processed using the same bioinformatic pipelines as described below. Duplicate HapMap dilution samples at 0.2, 0.5, 1, 2 and 5% VAF were called against a 0% 'pure' sample by MuTect2[8], Varscan2[9,10] and deepSNV[11,12]. The sensitivity and specificity of

**Fig. 3** Regional distribution of the mutations detected. **a–d** All detected somatic mutations within each brain region. SRMs or MRMs are indicated by differing colours, and the disease phenotype indicated by differing symbols. VAF for each mutation from the ACE platform is shown for each case. Only a single mutation in the cingulate was detected and therefore is not shown. The 95% confidence intervals (95% CI) are shown for the detected VAF using a normal approximation interval to account for any potential sampling bias corrected at each somatic mutation site incorporating the number of total reads and mutant reads. **e** Heat map showing the relative VAF as defined as the ratio between the lowest VAF compared to the VAF for the indicated sample for MRMs, highlighting that the VAF was consistently higher in the Medulla when sampled, and consistently low VAFs in the cerebellum. **f** Mutational signatures of each detected mutation in the study. The x-axis shows the 5′ and 3′ flanking base of each detected mutation, with the middle of the three alleles the reference allele that was mutated. The single base change for that allele is shown the column in which it is located (e.g., C > T, etc.), with each base mutation within a different column and depicted by a different colour. **g** Frequency of SRM and MRM mutations in specific genes seen in the 54 individual brains. **h** Summary of SRMs and MRMs observed in this study. Briefly, 62 somatic variants in 39 variant sites from 44 brain samples and 4 blood samples were detected by the ACE platform. Totally, 56 of these variants were also covered by the Haloplex[HS] platform, 22 somatic variants in 22 variant sites were SRMs, and 40 somatic variants in 17 variant sites were MRMs. 18 SRMs were observed in the brain samples and 14 were C > T substitutions

---

variant callers were determined using the following filtering formulae:

$$\text{Sensitivity} = \text{True Positive}/(\text{True Positive} + \text{False Negative})$$

$$\text{Specificity} = 1 - \text{False Positive Rate}$$

$$= 1 - \text{False Positive}/(\text{False Positive} + \text{True Negative}).$$

**Bioinformatic pipeline**. The bioinformatic pipeline is shown in Fig. 1e and in detail in Supplementary Figure 2. In overview, the primary calling pipeline included variants called by either MuTect2[8] or Varscan2[9,10] at a minimum VAF of 0.5%, before deepSNV[11,12] confirmed the presence of the detected variants in identified samples, and also confirmed that the VAF of detected somatic variants alleles in other brain regions was no different to the base error rate of other samples (where appropriate, Fig. 1e and Supplementary Fig. 2). The primary analysis was performed on the ACE data, and subsequently validated by the Haloplex[HS] platform.

ACE platform—SNVs and small indels were called using MuTect2 and Varscan2 Somatic Calling with the default parameters within the BED file for neurodegenerative and cancer 'control' genes. To detect SRMs we ran two callers on all possible sample pairs from one individual allocating each sample as the 'germline' or 'tissue' sample in turn. MRMs were called using Varscan single-sample Calling with VAF > 0.1% which may not be detected by paired calling, particularly in equivocal VAF between samples. We subsequently excluded variants: (1) with <1000 total read depth; (2) with <10 mutant reads; (3) <4 reads from either the forward or reverse strand; (4) those called as germline variants; (5) variants with a minor allele frequency (MAF) > 1% in 1000 genome project database[46], NHLBI ESP6500 (http://evs.gs.washington.edu/EVS/) or ExAC database[47] and (6) those within simple tandem repeats, segmental duplications, and microsatellites. All candidate mutations were subsequently annotated by ANNOVAR[48]. The same approach was used to analyse brain and blood combinations with a VAF > 0.5% based on spiked control sample data.

To ensure SRMs were truly focal, we utilised an additional caller to maximise the value of our large homogenous dataset ($n = 173$ samples at a mean coverage of 5374×) which ensured that identified SRMs and MRMs did differ from the base error rate seen in other samples. The deepSNV caller enabled us to build a separate error model for each base, and test whether the variant allele detected in the sample is greater than that expected from a beta-binomial distribution with an associated over dispersion factor which captures the observed degree of variation within the control samples. Testing this variant caller in spiked control samples (Fig. 1b) showed this caller to be the most specific caller of all those tested. To validate the focal variants, we ran deepSNV using other samples from the same individual as reference samples with a Benjamini-Hochberg corrected $P$ value for the number of samples tested. This ensured SRMs were: (1) identified by MuTect2 or VarScan2 Somatic Calling as present in only one region; (2) significantly different to all other samples within that individual (e.g., in the cerebellum and frontal cortex when detected in medulla); (3) significantly different from all other samples from all other individuals ($n = 169$ or $n = 170$) at the corrected threshold and (4) any other samples (e.g., frontal cortex and cerebellum) within that individual did not differ to all other samples from other individuals. MRMs were defined as variants with a VAF between 0.5% and 20% called in a single sample by Varscan2, and which were present in more than one region within the same individual. These putative MRMs were confirmed with deepSNV using the samples from other individuals as reference samples, and therefore MRMs were defined as follows: (1) as any variant identified by VarScan 2 single-sample calling to be present in more than one region; and which, (2) did again significantly differ to all other samples from other individuals at the corrected threshold.

Haloplex platform—To determine the accuracy of this approach, we validated identified variants using the Haloplex[HS] system data. In total 89.5% bases within the target region were covered above ×1000 and 94.3% bases were covered above ×500 (99.4% bases with coverage above ×1000 covered by ACE). In all, 62 variants were called initially on the ACE platform, and 56 (90.5%) were covered above ×500 on the Haloplex[HS] platform. The remaining six variants had VAF of >1% on the ACE platform in all but one case, with the remaining sample having a VAF of 0.53%. Given the high specificity of our approach using the ACE platform at the 0.5% threshold (98.25% Sensitivity and 100% Specificity) we considered these variants to be present with confidence. We also manually reviewed and confirmed the read alignments for all 62 somatic mutations detected on the ACE platform and 56 covered by Haloplex[HS] using IGV software (v.2.3)[49] confirming their presence.

Comparison of the detected VAF from each platform showed a strong correlation for the 56 variants ($r^2 = 0.953$, $P < 2.2e{-}16$) (Fig. 2b). These data indicate that our detection of somatic mutations was highly specific, and given that the DNA was independently amplified and sequenced, are highly unlikely to be due to amplification artefact.

Further QC steps: we ensured that all brain samples were from the same individual by performing identity by descent analysis based on called germline variants (see above) from both Genome Analysis Toolkit (GATK v3.5)[50–52] and Varscan2. Stringent QC filters were applied to remove poorly performing samples using PLINK v1.07 (http://pngu.mgh.harvard.edu/purcell/plink/)[53]. Single-nucleotide polymorphisms (SNPs) with a minor allele frequency of <40%, genotyping rate <95% or SNPs showing departure from Hardy-Weinberg equilibrium ($p < 1 \times 10^{-8}$) were excluded. Kinship coefficients were determined across the dataset as measured by the PI_HAT score in PLINK. Any sample with a PI_HAT score of >0.95 compared to any other sample from the same individual was removed and considered a duplication. This approach removed 6 samples as being incorrectly identified from 179 initial samples, leaving 173 samples within the final cohort.

The possibility of contamination from other samples was extremely unlikely because the MAF of all detected somatic alleles was <1% in reference databases (1000 genome, ExAc and ESP6500) in all cases, meaning that they are extremely rare in the population. All somatic variants were also cross-referenced with germline SNPs from cases within the Newcastle NBTR Brain Bank, in which 328 post-mortem brains have previously undergone Exome Sequencing[54]. No somatic variants in our current study were present in the heterozygous or homozygous state in that dataset.

**Mathematical analysis of mutation frequency**. Ultimately the brain, as with all organs, is derived from the single-cell zygote, so we modelled the process of brain development from fertilisation to account for potential somatic mutations that occurred before organogenesis. The branching model began with a single wild-type founder cell, and divided 37 times with zero probability of cell death, producing two daughters per division. Non-zero levels of cell death are explored in Supplementary Note 1. For the neurodegenerative disease genes, each base was modelled to have a constant probability of mutation per cell division. Using data for the case cohort, we used the approximate Bayesian computation rejection algorithm, with summary statistics of the data, to infer the somatic mutation rate per cell division during neurodevelopment based on this model. We used a broad uninformative prior for the mutation rate, spanning 4 orders of magnitude. With this, we simulated the development of the cell population in the brain as a branching process with 37 divisions required to generate the final cell population of neurons and non-neurons in approximately equal proportions[55]. By using one sample per individual from the approximate posterior distribution, we generalised the prevalence of pathological neuronal mutations to larger population sizes than those measured experimentally. See Supplementary Methods for further details.

**Quantification and statistical analysis**. All statistical calculations were performed as described in the text, with un-corrected $P$ values shown, and the corrected $P$ value provided for comparison where relevant.

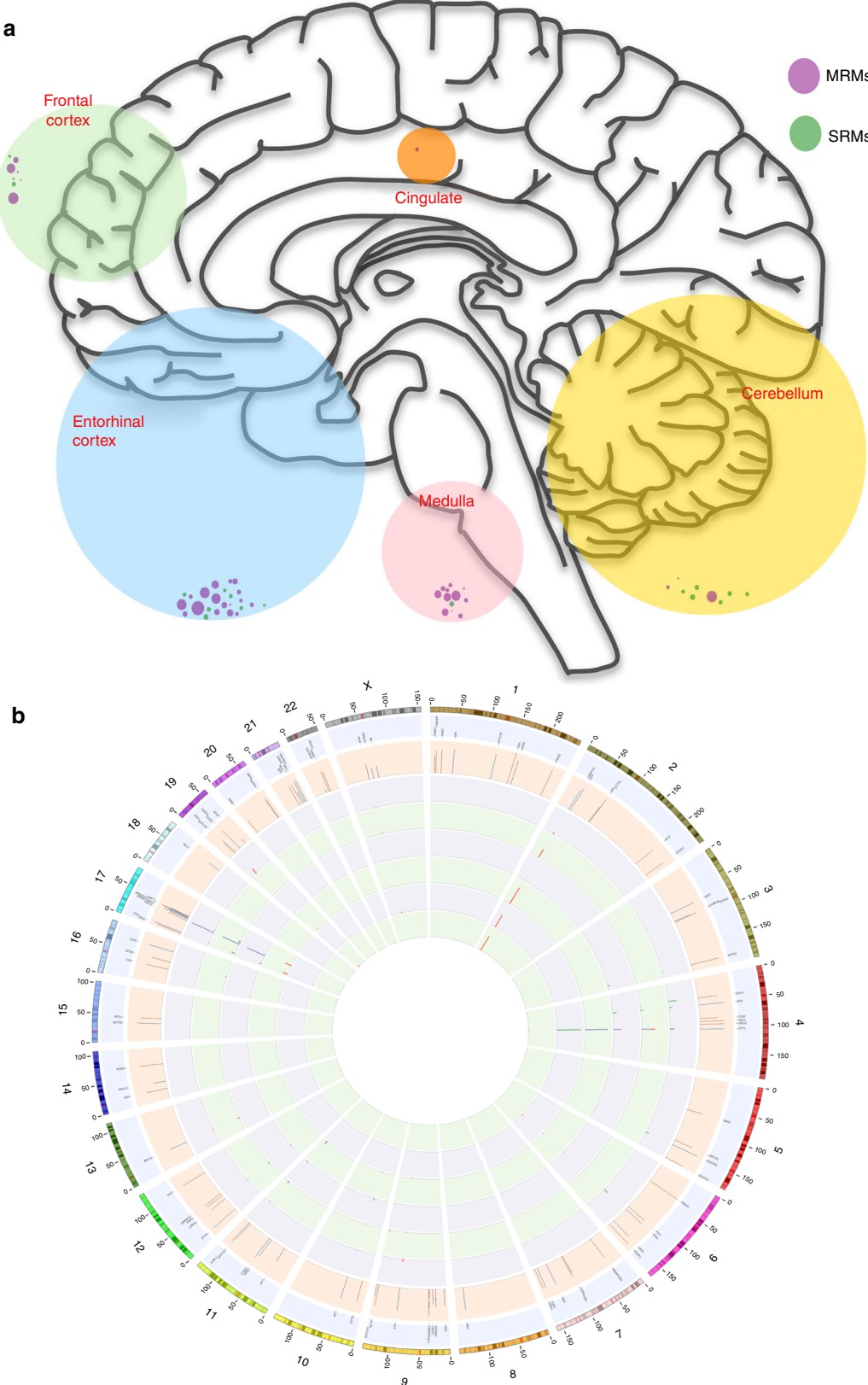

**Fig. 4** Relative proportions of clonal mutations within brain regions. **a** Neuroanatomical origin of the brain samples sequenced. Large circle radii are proportionate to the number of DNA molecules sequenced in each region. Smaller circles represent the proportion of SRMs (green) or MRM (purple) detected within that region in the 54 brains. **b** Circos plot showing the detected variants. Outer to inner: genomic positions on the autosomes and X-chromosome; 102 genes sequenced; mean sequencing depth for each gene; VAF in the cerebellum; VAF in the entorhinal cortex; VAF in the frontal cortex; VAF in the medulla; VAF in the cingulate; and VAF in the blood. Neurodegenerative disease gene mutations shown in red, and cancer gene mutations in blue

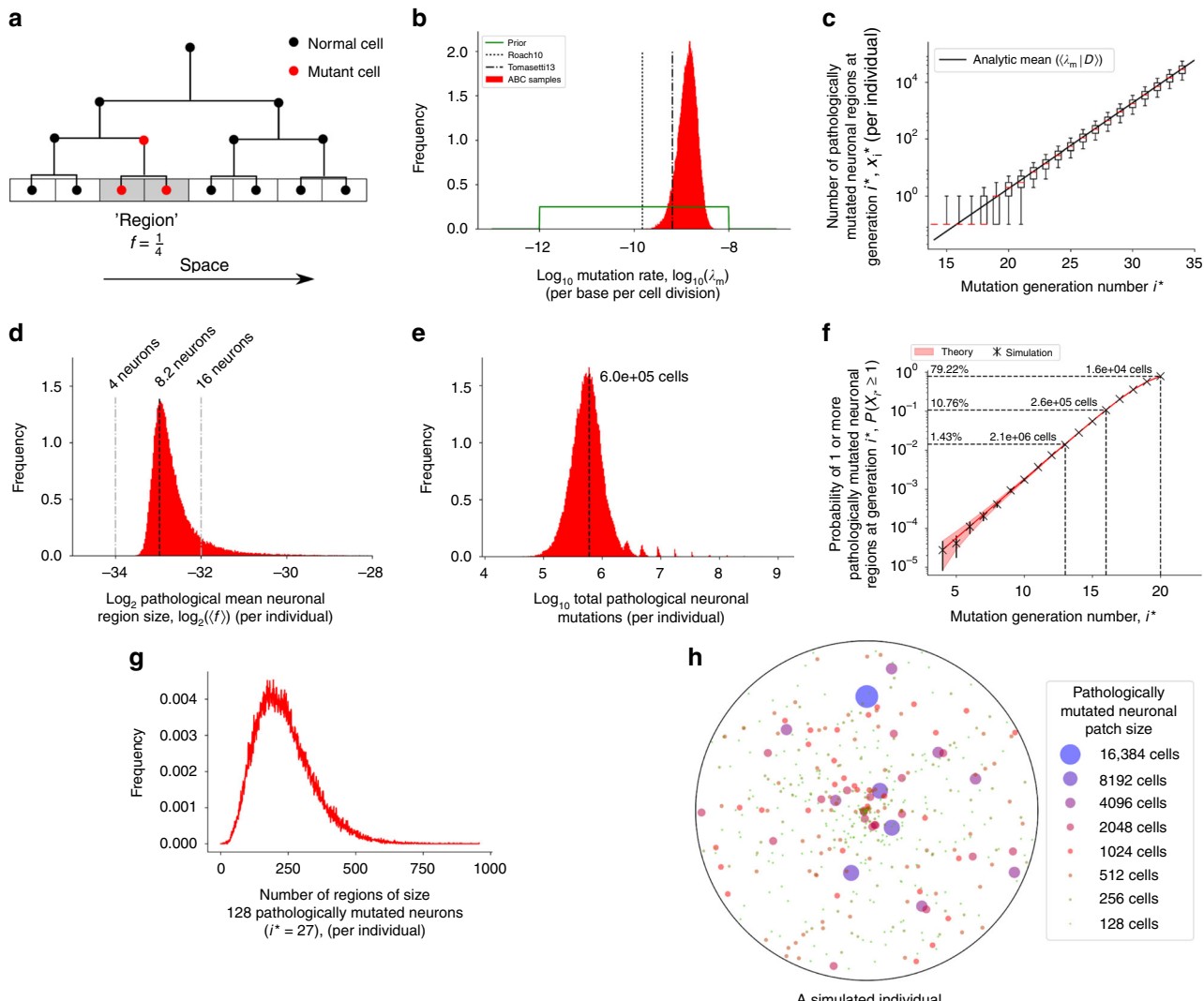

**Fig. 5** Estimates for the prevalence of somatic mutations. **a** Model of mutagenesis during brain development (Supplementary Methods; Supplementary Fig. 7). Red circle = cell containing a somatic mutation, black circle = cell with the wild-type allele. Development is modelled as a simple branching process. Every base was assumed to have a constant probability of mutation during replication. When a mutation arises in the lineage, its mutant daughters occupy a fraction $f$ of the adult brain, termed a 'region'. **b** Prior (green solid) and approximate posterior (red bars) for the mutation rate using approximate Bayesian computation, with literature values (black lines). **c–f** Using the approximate posterior samples for $\lambda_m$ we simulated brain development of the neuronal subpopulation (see Supplementary Methods). **c** Distribution of the number of mutated regions per individual, for each generation number ($i^*$), in any of the case genes associated with pathology. Boxplot whiskers show the 5th and 95th percentiles, median shown in red (when an individual has 0 mutations at generation $i^*$, the number of mutations is assigned to be 0.1, for representation on a log-scale). Showing the analytic mean of Eq. (16) in Supplementary Methods, where $\lambda_m$ is the approximate posterior mean (black solid line). **d** Distribution of the pathological mean relative region size across individuals. **e** Distribution of total number of pathological mutations across individuals. Multimodality is induced by the largest pathological region (Supplementary Fig. 7). **f** Fraction of simulated human individuals with at least 1 pathologically mutated region seeded at generation $i^*$. Comparison to theory shown in red, mean defined in Eq. (20) and standard deviation Eq. (21). (Error bars are a Bernoulli error model). **g** Frequency distribution over individuals of pathologically mutated regions of size 128 neurons. **h** Representation of the size and frequency distribution of pathologically mutated regions (seeded between $20 \leq i^* \leq 27$) in an individual human brain. Whole brain area (black circle) is not to scale with the mutated regions (coloured circles). Data used in the model inference was derived from neurodegenerative patients—but no significant difference in mutation prevalence of case genes between control and case patients were observed

All samples throughout the study were given an anonymous sample ID and were randomly assigned to sequencing runs. Sample sizes were performed as calculated below. Biological replicates from brain tissue samples were performed on an orthogonal sequencing platform as described in the text after biological replicates were used for spiked HapMap control data revealing high concordance.

**Confidence intervals for observed variant AFs**. In order to account for potential effects of sampling error in the determination of the VAF on the ACE platform, we calculated the normal approximation interval based upon the total read depth at

each somatic allele site, and the number of variant reads observed.

$$\frac{1}{n}\left(n_s \pm z\sqrt{\frac{1}{n}n_s n_f}\right)$$

Where $n$ = the total number of reads, $s$ = the number of variant reads, and $f$ = the number of reference allele reads.

**Power calculations.** To calculate the number of samples required from of any neurodegenerative disease (e.g., LB disease or AD) and controls that would give 90% power with 95% confidence (assuming a type 1 error rate of 5%), to detect double the frequency of somatic mutations in case genes (26%) in any disease cohort compared to controls (13%) (assuming equal matching), we used the following equation:

$$n_A = kn_B \text{ and } n_B = \left( \frac{p_A(1-p_A)}{k} + p_B(1-p_B) \right) \left( \frac{z_{1-a/2} + z_{1-\beta}}{p_A - p_B} \right)^2$$

$$z = \frac{(p_A - p_B)}{\sqrt{\frac{p_A(1-p_A)}{n_A} + \frac{p_B(1-p_B)}{n_B}}}$$

where $k$ is the matching ratio (assuming a 1:1 ratio between cases and controls), $\alpha$ is the Type 1 error and $\beta$ is the type II error.

## Data availability

Sequencing data—aligned bam files of 179 samples using ACE technology are deposited in NCBI Sequence Read Archive (SRA) under accession code SRP159015.

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

## Acknowledgements

MJK is a Wellcome Trust Clinical Research Training Fellow. JA is supported by Biotechnology and Biological Sciences Research Council (BB/J014575/1). N.S.J. is supported by the Engineering and Physical Sciences Research Council (EP/N014529/1). P.F.C. is a Wellcome Trust Senior Fellow in Clinical Science (101876/Z/13/Z), and a UK NIHR Senior Investigator, who receives support from the Medical Research Council Mitochondrial Biology Unit (MC_UP_1501/2), the Medical Research Council (UK) Centre for Translational Muscle Disease research (G0601943), EU FP7 TIRCON, and the National Institute for Health Research (NIHR) Biomedical Research Centre based at Cambridge University Hospitals NHS Foundation Trust and the University of Cambridge. The views expressed are those of the author(s) and not necessarily those of the NHS, the NIHR or the Department of Health.

## Author contributions

M.J.K. designed laboratory and bioinformatics experiments, performed experiments, analysed the data, performed statistical analyses, and wrote the first draft of the paper; W.W. designed and performed the bioinformatic and statistical analysis, and contributed to the manuscript; J.A. designed mathematical analysis of sequencing data, performed mathematical experiments, performed statistical analysis and wrote the Supplementary Methods and Supplementary Notes; L.W. performed experiments; Jv.d.A. advised on brain development; J.C. performed experiments; I.W. designed experiments; M.B. assisted in performing bioinformatic analyses; J.B. assisted in study design, analysed and provided sequencing data; J.W., R.C., C.H., G.B. and S.L. provided technical expertise and assistance with the sequencing. C.M.M. assisted in case selection and performed pathological examinations; N.S.J. designed mathematical analysis of sequencing data, performed mathematical experiments and wrote the Supplementary Methods and Supplementary Notes; J.A.A. assisted in case selection and performed pathological examinations; P.F.C. supervised the project, assisting in the study design and interpretation. All authors contributed to the manuscript.

## Additional information

**Competing interests:** The authors declare no competing interests.

