## [Peer Review File · Nature Communications]

Reviewer #1 (Remarks to the Author):

In the presented manuscript, Keogh et al. conducted panel sequencing of exons of 102 genes (56 are neurodegeneration associated and 46 are cancer associated) in 173 regions of 54 brains. The aim of the study was to discover somatic mutations in the brain and analyze how they could be relevant to neurodegenerative diseases.

After spending substantial efforts on computational workflow and confirming their approach (through experimental sample spike-in and experimental, though not quite orthogonal, validation) they discovered 27 somatic mutations.

Next, they conducted mathematical modeling to deduce mutation rates in the brain and estimate frequency of regions with pathologically mutated cells in an individual brain.

While I find the first half of the study — discovering mutations — quite convincing, I find the second part — modeling and projections — as highly superficial. I overall think that their model qualitatively correct, but I doubt it can be used to make qualitative statements because of unrealistic assumptions and crudely estimated parameters in the model.

Specifically:

* They assume that neurodevelopment begins with a single founder cell

It seems to me that the larger the founder population, the fewer and smaller in size pathologically mutated zones can exist in a brain. Isn't that true? How would the number of zones in the abstract change if there were 100 or 1000 founding cells?

* They model brain development as deterministic branching process

The authors and I agree that there are symmetric and asymmetric stages of neurogenesis. In the supplement, the authors use 1.37×10^{11} as the total number of cells in brain and assume that all of them were created through symmetric division or doublings. One can easily estimate that realistically ~98% or more of these cells are generated through asymmetric cell division (i.e., asymmetric cell division continues for about 12 weeks during fetal development, with cell cycle between 24-48 hours, leading to a minimum of 42 divisions for each progenitor). Thus, the number of doubling is 32 or 31 but not 37 as the authors assume in their model. Actually, it must be even smaller, because of multiple founding cells (see my comments to the previous issue).

The authors admit in the text that their model is

“... a crude approximation of real neurodevelopment, neglecting subtleties relating to the precise mechanisms of cell death; asymmetric cell division in the later stages; the existence of neuronal progenitors and cellular migration for example (see Supporting Information for discussion of these points).”

and I think it would be fair to the reader that they also report in the text how their estimates for quantity and the size of the pathologically mutated zone would change if they used 30 and 25 doublings.

In my view the following:

“However, if neurodevelopment is topologically similar, but not precisely equivalent to a deterministic branching process, then we expect islands of pathologically mutated cells to exist.”

is as far as they can go with the model.

Thus, I think that quantitative results from the model should not be stated in the abstract.

Reviewer #2 (Remarks to the Author):

Keogh et al performed deep sequencing for a panel of 102 genes, encompassing candidate genes for neurodegenerative disorders and cancer related genes, across multiple regions of Alzheimer disease (AD), Parkinson disease with Lewy body (LB) and age matched control brains. Somatic mutations were detected and validated in approximately half the brains, some present in multiple brain regions. There was no significant difference in prevalence for mutations in neurodegenerative disease genes vs cancer genes across the brains, and no apparent significant difference in their distribution between patients and controls. The authors estimate this to a lack of power due to insufficient case number. Using a number of assumptions, they subsequently mathematically model

brain development and use their data to estimate that approximately 10% of individuals have roughly 50-60,000 mutations distributed in 75-480 foci of various size throughout the brain. The authors should be commended for the accuracy of their sensitivity and specificity calculations, the high average sequencing depth and the validation approach. The results are original and of considerable interest and could stimulate further research, but the manuscript should be revised as suggested below.

1. Single region mutations in neurodegenerative disease genes and cancer genes were equally likely to occur in the brains examined, and there was no difference in the proportion in mutations in neurodegenerative disease genes between the patient groups (AD, LB, and controls). For multiple regional mutation, only 1 out of 39 occurred in a neurodegenerative disease gene. Hence, even though they chose to model only neurodegenerative disease gene mutations, the claim made in the abstract and in multiple areas of the paper that focal pathological mutations in neurodegenerative genes are likely to be present in all human brains is overreaching. For a broadly oriented Journal, the abstract should represent a more balanced overview of the major results, not just the conclusion of the mathematical modeling of neurodegenerative disease gene mutations.
2. The abstract further claims that these focal mutations could lead to the accumulation of misfolded proteins; however, this conclusion has no basis in the actual results. The authors should estimate the focal accumulation of such proteins to report this in the abstract. Were the cancer-related mutations also focal, as it seems likely?
3. The work is presented in a sometime confusing way. For example, the introduction concludes "Here we took a complementary approach, harnessing ultra-high depth sequencing to survey the somatic mutational landscape across different brain regions". However, their work uses ultradeep sequencing to survey a limited number of candidate mutations, rather than assessing the whole mutational landscape. The words "mutational landscape" should be changed.
4. From Fig. 3 it seems that the entorhinal cortex had a higher number of mutations overall. Was this statistically significant?
5. To determine the focal mutation rate, how was the cell cycle time approximated? Also, does the model take into account mutations that occur in post-mitotic cells? The mutation rate during neurogenesis has been estimated to be considerably higher than here (Bae et al, 2018).
6. The authors admit themselves that because of the multiple assumptions they must make (number of cell divisions generating the brain, lack of asymmetric divisions in brain development, strict spatial proximity of cell lineages, lack of migration and cell death) their model is likely strongly oversimplified and quite possibly biased. For example, to calculate the distribution and number of mutated regions per generation and individual (Fig. 3c) the time when these brain regions are formed in development should be imputed in the model, using sharing of mutations between brain regions. This is because the different brain regions, and the brain itself, do not arise from a single cell, but from a number of founder cells.
7. Brain regions are of different size and cells are motile and cross brain region boundaries before the region boundaries are biologically implemented by cell adhesion boundaries.

8. The fact that some of the high frequency mutations, which are implied to occur very early in development, do not seem to be shared between brain regions suggests that other factors are accountable for their frequency (i.e., selective pressure). Another explanation, albeit unlikely, is that human brain regions obey different compartmentalization rules than those of other vertebrates. Either way the model is not sufficiently explanatory.

Typos:

multi region mutations, SRMs (page 4, line 90)

Medella (Fig 3C)

brain 54 (control) (page 8 line 240)

Reviewer #3 (Remarks to the Author):

The authors set out to test whether somatic mutations in brain cells could be causal of neurodegenerative disorders. This is a very interesting hypothesis. The authors used deep sequencing to identify somatic mutation, followed by a theoretical modelling showing that regions of pathological mutations acquired through development are likely mechanisms for neurodegenerative disorders. The molecular design of the study is good and the theoretical model seems to produce results that overlap with previous reports. However, there are several troubling issues.

First, the writing style of the method section makes it really hard to keep the track of how consistent the reported numbers are. For example on page 5 the authors state that they detected 62 somatic variants in total. However, in the following sections they report 39 as the total number of variants! I might be missing something, but if so this highlights the lack of clarity. Moreover, even when the numbers do add up, one has to search through the report to find where the total quoted number came from. For example, in section "Single regional mutations (SRMs)" page 5 line 134, the authors state that there were 15 out of 18 C>T substitutions, it is hard to work out where the 18 came from. I had to count previously reported numbers to figure out that this was referring to lines 117-118 on page 5. This type of reporting is pretty frustrating for the reader and makes it hard to follow. It could be beneficial to provide a summary table where all total numbers are summarised helping the reader to keep a track of what is going on.

Minor note, there is a typo in a total number of brains, page 5 line 116. The authors state that the total number (after QC) was 44 brain samples, but I think this should be 54.

The mathematical model presented has good theory behind it, however I do not understand why the modelling was limited to 40 individuals where the study sample was clearly 54? The authors clearly report that the mutation were found in controls as well (6 out of 14), therefore if the justification was based on the sample size, then I believe it should be equal to the sample size. It perhaps is a trivial point but clearly 40 is not the sample size of the current study.

Finally, although the findings are intriguing, due to a very small sample size and the lack of independent replication, the interpretation of them should be much more cautious. Particularly when explicitly generalising to the population. At times the report feels more appropriate as a case study. Some conclusions are based on a sample $n=1$ running a very realistic danger of over interpretation. In addition, of the 27 brains that have shown somatic mutations, six were healthy making the pathogenesis hypothesis less clear cut.

Response to the reviewers

Reviewer 1

Reviewer: While I find the first half of the study — discovering mutations — quite convincing, I find the second part — modeling and projections — as highly superficial. I overall think that I their model qualitatively correct, but I doubt it can be used to make qualitative statements because of unrealistic assumptions and crudely estimated parameters in the model.

Response: Whilst we agree that the model presented is crude, and only intended to make preliminary statements about the prevalence of somatic mosaicism in the brain in the general population, we have now shown that this simple model is robust to several modelling perturbations, including those suggested by the reviewer (Sections S2, S3 of the updated Mathematical supplement and Supplemental Figures 9 & 10).

At its base, this robustness arises from the following observation. If we assume that the mutation rate of DNA is on the order of $10^{-0w}-10^{-0}$ mutations per base per cell division (as found in our work in Fig. 5b, and others), given unbiased, spatially proximal, replication of daughter cells once the brain consists of approximately 10^6 cells, a simple order-of-magnitude estimate (see updated Mathematical Supplement - Section S4) suggests that every individual is expected to possess 1 pathologically mutated focal mutation consisting of approximately 10^4-10^5 cells. This argument is independent of the neurodevelopmental mechanism prior to the brain consisting of 10^6 cells. The argument also only relies on unbiased, spatially proximal, replication: the precise details of neurodevelopment are also unimportant given these two key assumptions. Many of the modelling suggestions which have been helpfully suggested by the reviewers possess these two properties. We have since added this argument to the Main Text.

Following the reviewer's recommendations, we have rephrased the Main Text to emphasise that the model is principally for demonstrating the qualitative observation of somatic mosaicism. We hope that the new evidence we present shows that the model is a useful means of performing preliminary order-of-magnitude estimations. We thank the reviewer for drawing attention to these issues, and in making several interesting suggestions to test the model's robustness which we address below.

Reviewer: Specifically: They assume that neurodevelopment begins with a single founder cell. It seems to me that the larger the founder population, the fewer and smaller in size pathologically mutated zones can exist in a brain. Isn't that true? How would the number of zones in the abstract change if there were 100 or 1000 founding cells?

Response: We have now investigated the issue of multiple founder cells. Whilst the reviewer's intuition is correct, the effect size is very small. In our original model, mutations which affect the first 1000 cells only occur with probability $\sim 1/1000$ or less (see Fig 5F). Thus, their contribution to the total number of different mutations per individual is small, and the effect on the average mutant zone size is negligible, since there are exponentially many more cells from later generations. Although earlier mutations (ie before 1000 cells are formed) would lead to larger 'mutation zones' in the brain, these are very rare in the population, and thus do not have a significant impact on the overall behaviour of the model when we consider many human brains. As a consequence, the prediction of e.g. $\sim 10\%$ of individuals harbouring 10^5 spatially-contiguous pathologically mutated neurons is unchanged when we alter the model along the lines the reviewer suggests (one would require approximately 10^4 founder cells (as per Fig 5F at the 1% level, consisting of 2^{13} cells), or more, for founder effects to start significantly impacting our qualitative conclusions). We have revised section S2 of the Mathematical Supplement to present these findings.

Reviewer: They model brain development as deterministic branching process. The authors and I agree that there are symmetric and asymmetric stages of neurogenesis. In the supplement, the authors use 1.37×10^{11} as the total number of cells in brain and assume that all of them were

created through symmetric division or doublings. One can easily estimate that realistically ~98% or more of these cells are generated through asymmetric cell division (i.e., asymmetric cell division continues for about 12 weeks during fetal development, with cell cycle between 24-48 hours, leading to a minimum of 42 divisions for each progenitor). Thus, the number of doublings is 32 or 31 but not 37 as the authors assume in their model. Actually, it must be even smaller, because of multiple founding cells (see my comments to the previous issue).

Response: We have now investigated the issue of asymmetric division. We used a liberal estimate of 82 asymmetric divisions per progenitor, and 30 symmetric divisions beforehand. We find that the mean mutant region size changes from ~8 neurons to ~31 neurons, and the total number of mutations per individual brain changes from 6×10^5 cells to 2×10^6 . The prediction of e.g. ~10% of individuals harbouring 10^5 spatially-contiguous pathologically mutated neurons remains unchanged when we change the model in this way. We have revised section S3 of the Mathematical Supplement to present these findings.

Combining the progenitor pool model and the asymmetric division model is considered in the next reviewer response.

Reviewer: The authors admit in the text that their model is: "... a crude approximation of real neurodevelopment, neglecting subtleties relating to the precise mechanisms of cell death; asymmetric cell division in the later stages; the existence of neuronal progenitors and cellular migration for example (see Supporting Information for discussion of these points)."...and I think it would be fair to the reader that they also report in the text how their estimates for quantity and the size of the pathologically mutated zone would change if they used 30 and 25 doublings.

Response: We have now altered the main text to point the reader to the Mathematical Supplement where these issues are explored in depth. It is worth, however, re-emphasizing that all of our quantitative observations were robust to within an order of magnitude across all of the perturbations we made to the simple deterministic branching model, and intuition for this is provided in Section S4 of the Mathematical Supplement.

Mixing both the founder effect model, and the asymmetric growth model, is expected to have minimal effect on our predictions pertaining to the propensity of somatic mosaicism in the general population, since this hybrid model obeys conditions (1) and (2) in Section S4 of the Mathematical Supplement.

Reviewer: In my view the following: "However, if neurodevelopment is topologically similar, but not precisely equivalent to a deterministic branching process, then we expect islands of pathologically mutated cells to exist."...is as far as they can go with the model. Thus, I think that quantitative results from the model should not be stated in the abstract.

Response: We agree, and have altered the text to reflect this, including the abstract.

Reviewer 2

Reviewer: The results are original and of considerable interest and could stimulate further research, but the manuscript should be revised as suggested below.

Single region mutations in neurodegenerative disease genes and cancer genes were equally likely to occur in the brains examined, and there was no difference in the proportion in mutations in neurodegenerative disease genes between the patient groups (AD, LB, and controls). For multiple regional mutation, only 1 out of 39 occurred in a neurodegenerative disease gene. Hence, even though they chose to model only neurodegenerative disease gene mutations, the claim made in the abstract and in multiple areas of the paper that focal pathological mutations in neurodegenerative genes are likely to be present in all human brains is overreaching. For a broadly oriented Journal, the abstract should represent a more balanced overview of the major results, not just the conclusion of the mathematical modeling of neurodegenerative disease gene

mutations.

Response: We agree with the reviewer, and have redrafted the manuscript, reducing the emphasis on neurodegenerative disease genes. This includes the following: (1) we have removed reference to neurodegenerative disease genes in the abstract; (2) in the final paragraph of the discussion we refer to neurodegenerative disease genes in the context of a more general process across the whole genome.

Reviewer: The abstract further claims that these focal mutations could lead to the accumulation of misfolded proteins; however, this conclusion has no basis in the actual results. The authors should estimate the focal accumulation of such proteins to report this in the abstract. Were the cancer-related mutations also focal, as it seems likely?

Response: The reviewer is correct on both counts, some cancer-related mutations were also focal, and we have no experimental evidence that the focal mutations could lead to the accumulation of mis-folded proteins. We have therefore deleted reference to misfolded proteins in the abstract. In addition, we have added a caveat that in the discussion, that we currently have no direct evidence that the focal mutations contribute to neurodegenerative disease.

Reviewer: The work is presented in a sometime confusing way. For example, the introduction concludes “Here we took a complementary approach, harnessing ultra-high depth sequencing to survey the somatic mutational landscape across different brain regions”. However, their work uses ultradeep sequencing to survey a limited number of candidate mutations, rather than assessing the whole mutational landscape. The words “mutational landscape” should be changed.

Response: We agree with the reviewer and have deleted ‘mutational landscape’, simply referring to mutations.

Reviewer: From Fig. 3 it seems that the entorhinal cortex had a higher number of mutations overall. Was this statistically significant?

Response: There was no statistically significant difference in either: (1) The proportion of entorhinal cortex samples having somatic mutations compared to the rest of brain regions (Fisher exact test $P = 0.21$); nor (2) The mean number of somatic mutations in entorhinal cortex compared to the other of brain regions (Wilcoxon rank sum test $P = 0.102$)

Reviewer: To determine the focal mutation rate, how was the cell cycle time approximated? Also, does the model takes into account mutations that occur in post-mitotic cells? The mutation rate during neurogenesis has been estimated to be considerably higher than here (Bae et al, 2018).

Response: In our original model presented in the Main Text, we do not need a cell cycle time to estimate the focal mutation rate, because the model is a discrete-time model with non-overlapping generations. One only requires the total number of generations. Our Bayesian approach then involves “guessing” the value of the mutation rate, and through simulation, if that mutation rate gives rise to an in-silico dataset which “looks like” the data, we accept this guess as a plausible mutation rate. No notion of physical time is required to perform this kind of inference.

Our approach does not take into account mutations which occur in post-mitotic cells. Whilst we agree with the reviewer that mutations e.g. induced by heavy transcription of particular genes could influence the distribution of inferred mutation rates, we have erred on the side of parsimony and neglected this aspect of mutagenesis in our modelling efforts. We note that our inferred mutation rate could conceivably have been inflated by post-mitotic mutagenesis; however, since our inferred distribution of mutation rates is broadly compatible with those found in the literature (between 10^{-10} and 10^{-9} mutations per base per cell division), this provides an indication that our modelling approach is appropriate for order-of-magnitude estimation.

Bae et al (2018) reported 1.3 mutations per division per cell. Based on our data, we estimate the

mutation rate at 4.8×10^{-10} to 2.99×10^{-9} mutations per division per cell. Scaling up our values across the whole genome (3.2×10^9 base pairs per haploid genome), our estimation corresponds to 6.1-38 mutations per division per cell. Thus, our results are within one order of magnitude of the results of Bae et al (2018). This is remarkable, given that we used a completely different experimental approach based on adult brain samples. Interestingly, Bae et al (2018) also reported 8.6 (95% CI 1.6-20) *de novo* mutations per division per cell when they cultured single cells harvested from the brain. Their confidence intervals using this technique overlap with ours, further validating our findings.

Reviewer: The authors admit themselves that because of the multiple assumptions they must make (number of cell divisions generating the brain, lack of asymmetric divisions in brain development, strict spatial proximity of cell lineages, lack of migration and cell death) their model is likely strongly oversimplified and quite possibly biased. For example, to calculate the distribution and number of mutated regions per generation and individual (Fig. 3c) the time when these brain regions are formed in development should be imputed in the model, using sharing of mutations between brain regions. This is because the different brain regions, and the brain itself, do not arise from a single cell, but from a number of founder cells.

Response: We have since investigated the effect of multiple founder cells in neurodevelopment, see section S2 of the Mathematical Supplement. Assuming that neurodevelopment begins with ~1000 cells or fewer has very little effect on the quantitative predictions of the original model. This is because mutations which affect the first 1000 cells in the original model were rare (about 1:1000, see Fig 5F and our response to reviewer 1).

Reviewer: Brain regions are of different size and cells are motile and cross brain region boundaries before the region boundaries are biologically implemented by cell adhesion boundaries.

Response: In the revised section S4 of the Mathematical Supplement we discuss that, indeed, neurodevelopment shows aspects of diffusivity, which serves to degrade spatial correlations throughout the brain. The greater the extent of cellular diffusion through neurodevelopment, the less applicable spatially embedded tree-like models of neurodevelopment are. We draw the attention of the reviewer to section S5 of the Mathematical Supplement, where we provide an entirely orthogonal model which is more appropriate if diffusion is strong. Surprisingly, we found that the total number of pathologically mutated cells is comparable to the original model.

Reviewer: The fact that some of the high frequency mutations, which are implied to occur very early in development, do not seem to be shared between brain regions suggests that other factors are accountable for their frequency (i.e., selective pressure). Another explanation, albeit unlikely, is that human brain regions obey different compartmentalization rules than those of other vertebrates. Either way the model is not sufficiently explanatory.

Response: It is plausible for a “high frequency” mutation, affecting e.g. ~10% of a ~3000 cell sample, to occur relatively late within development. Applying the formula $f = 2^{-(i^*+2)}$, where f is the fraction of the brain affected by a mutation and i^* is the mutation generation number (and an entire brain consists of 37 generations), yields $i^* = -\log_2(0.1 \cdot 3000 / 2^{37}) - 2 = 27$. From Fig 5G, we see that an individual is expected to possess ~250 pathologically mutated regions under our simple model. We would argue that the existence of such “high frequency” mutations is not direct evidence for the requirement of a more complex model, or for the need to invoke a selective pressure, and it is not obvious that our model is insufficiently explanatory given the robustness exercises we have performed. This is not to say that selective pressures/compartmentalization effects do not exist. However, they do not appear to be necessary to account for this data set. We have therefore erred on the side of parsimony for the purpose of this preliminary model.

Reviewer: Typos:
multi region mutations, SRMs (page 4, line 90)
Medella (Fig 3C)

brain 54 (control) (page 8 line 240)

Response: We have corrected all of the typos highlighted by the reviewer.

Reviewer 3

Reviewer: First, the writing style of the method section makes it really hard to keep the track of how consistent the reported numbers are. For example on page 5 the authors state that they detected 62 somatic variants in total. However, in the following sections they report 39 as the total number of variants! I might be missing something, but if so this highlights the lack of clarity. Moreover, even when the numbers do add up, one has to search through the report to find where the total quoted number came from. For example, in section “Single regional mutations (SRMs)” page 5 line 134, the authors state that there were 15 out of 18 C>T substitutions, it is hard to work out where the 18 came from. I had to count previously reported numbers to figure out that this was referring to lines 117-118 on page 5. This type of reporting is pretty frustrating for the reader and makes it hard to follow. It could be beneficial to provide a summary table where all total numbers are summarised helping the reader to keep a track of what is going on.

Response: The reviewer raises a very important point. To address this we have produced a new figure (Fig.3h) which summarises the samples we studied and the mutations we detected. We have also added the following text to the manuscript:

Eighteen somatic mutations (56.4% of the total 39 detected variants) were present in only one brain region within an individual and 4 somatic variants in only one blood sample (Figs. 2c, d, 3a-e, h, Supplementary Figures 3-5 and Supplementary Tables 5 and 6).

We also corrected an error in the previous text “The majority of SRMs in brain were C>T substitutions (n=15/18, 83.3%)” as the following:

The majority of SRMs in brain were C>T substitutions (n=14/18, 77.8%).

Reviewer: Minor note, there is a typo in a total number of brains, page 5 line 116. The authors state that the total number (after QC) was 44 brain samples, but I think this should be 54.

Response: We apologise for the confusion here. The number of 44 (brain samples) was correct. After QC, we studied 173 brain samples and 6 blood samples from 54 individuals. 62 somatic variants were detected in 44 brain samples and 4 blood samples. We have clarified the numbers in the new figure (Fig.3 h) and changed the sentence in the previous text “the ACE platform detected 62 somatic variants **from** 44 brain samples and 4 blood samples” to the following:

*..the ACE platform detected 62 somatic variants **in** 44 brain samples and 4 blood samples.*

Reviewer: The mathematical model presented has good theory behind it, however I do not understand why the modelling was limited to 40 individuals where the study sample was clearly 54? The authors clearly report that the mutation were found in controls as well (6 out of 14), therefore if the justification was based on the sample size, then I believe it should be equal to the sample size. It perhaps is a trivial point but clearly 40 is not the sample size of the current study.

Response: Our data is split into patients (affected by neurodegenerative diseases) and healthy control brains. Although there is insufficient evidence to reject the null hypothesis that both patients and controls have different propensities to single regional mutation, we decided to restrict ourselves to just the patient data. This is because an inability to reject the null hypothesis from insufficient quantities of data does not mean support for the null hypothesis. Since the control data is in the minority, instead of invoking a more complex model where we allow for different mutation rates for patient and control groups, we took a more parsimonious route and decided to focus on the group with the largest amount of data, namely the patient group. We have clarified this point in the Mathematical Supplement.

Reviewer: Finally, although the findings are intriguing, due to a very small sample size and the lack of independent replication, the interpretation of them should be much more cautious. Particularly when explicitly generalising to the population. At times the report feels more appropriate as a case study. Some conclusions are based on a sample n=1 running a very realistic danger of over interpretation. In addition, of the 27 brains that have shown somatic mutations, six were healthy making the pathogenesis hypothesis less clear cut.

Response: We agree with the reviewer. In the revised manuscript we have 'toned down' our extrapolations and conclusions, and make it clear that our findings only raise the possibility that the mutations might be important for disease.

Reviewer #1 (Remarks to the Author):

The authors have, more or less, addressed my comments.

I disagree with the authors' interpretation that "... mutations which affect the first 1000 cells only occur with probability $\sim 1/1000$ or less (see Fig 5F). Thus, their contribution to the total number of different mutations per individual is small ...". In my view, it is not that their contribution is small but rather that, if there is a pathogenic mutation in one of those 1000 cells, a person is likely to exhibit a disease phenotype and won't be considered normal. I would say that formally mutations in the 1000 progenitors are not pathogenic by definition. But this is a minor point.

A serious concern is the following. The authors claim that, when using 1024 progenitors as a model for the brain, the number of pathological regions does not change. That claim strongly contradicts the logic of the model. The sizes of regions with pathological mutations should become inversely smaller with an increasing number of progenitors. If there are 1024 progenitors instead of 1, then each progenitor has to experience 10 symmetrical divisions fewer to generate a brain of the same size. Thus, the sizes of pathological regions must get smaller, while the total number of mutations is roughly the same. Can the authors comment on this?

Reviewer #2 (Remarks to the Author):

The authors have satisfactorily addressed my concerns.

Reviewer #3 (Remarks to the Author):

The authors addressed my concerns satisfactorily

Response to reviewer 1

Reviewer: I disagree with the authors' interpretation that "... mutations which affect the first 1000 cells only occur with probability $\sim 1/1000$ or less (see Fig 5F). Thus, their contribution to the total number of different mutations per individual is small" In my view, it is not that their contribution is small but rather that, if there is a pathogenic mutation in one of those 1000 cells, a person is likely to exhibit a disease phenotype and won't be considered normal. I would say that formally mutations in the 1000 progenitors are not pathogenic by definition. But this is a minor point.

Response: We agree that the reviewer, but unfortunately our study was not designed to address this issue.

Reviewer: A serious concern is the following. The authors claim that, when using 1024 progenitors as a model for the brain, the number of pathological regions does not change. That claim strongly contradicts the logic of the model. The sizes of regions with pathological mutations should become inversely smaller with an increasing number of progenitors. If there are 1024 progenitors instead of 1, then each progenitor has to experience 10 symmetrical divisions fewer to generate a brain of the same size. Thus, the sizes of pathological regions must get smaller, while the total number of mutations is roughly the same. Can the authors comment on this?

Response: The reviewer seems to be referring two separate issues here, (1) the number of regions affected by a mutation; and, (2) the size of the regions affected by a mutation.

Our basic model indicates that – providing the total number of mutation events is the same – the number of regions affected by a mutation does not change, irrespective of when they occurred. However, the size of the regions affected by a mutation will vary, depending on when the mutation occurred, with earlier mutations affecting a greater region of the brain.

Take, for instance, the extreme limit, where the brain consisting of a massive pool of progenitor cells undergo 1 symmetric division to form the adult brain. Neglecting lag subtleties (SF7A & Eq(4)), a mutation in this single division will affect no more than 2 cells. Contrast this to the case where there is 1 progenitor: a mutation in the original progenitor can essentially affect the entire brain (so 2^{36} cells). So indeed, the number of progenitors limits the size of the brain that can be affected by somatic mutations – but in both cases, the total number of mutations present in the brain is the same.

Given that the observed mutation rate is low, the likelihood of having a mutation in the first 10 symmetrical divisions is very low. As a result, the vast majority of brains will not have a mutation acquired at this time of development. In the original model with 1 progenitor, the low mutation rate means that only 1/1000 simulated brains acquired a mutation in the first 10 divisions (Fig 5F, corresponding to 1024 cells). Once the original model reaches generation 10, if there has not been a mutation by this point, the original model behaves the same as the progenitor model (see the figure below). This is the case for $\sim 99.9\%$ of individuals. Thus, the number of pathological regions in both models, and the size of each corresponding region, is the same for almost all individuals.

To summarise:

- In the original model, if a mutation has not occurred by generation 10, the progenitor model is behaves like the original model (see the figure above).
- In the original model, $\sim 1/1000$ individuals or fewer experience a mutation in the first 10 divisions.
- So, for $\sim 99.9\%$ of individuals, the progenitor model is equivalent to the original model
- These 99.9% are in such a vast majority that they wash out the statistical signal from the 0.1% who are differentially affected between the two models, as shown in SF9.

We have added this summary to the Mathematical Supplement.

Reviewer #1 (Remarks to the Author):

Addressed